# Unsupervised Anomaly Detection with Distillated Teacher-Student Network Ensemble

**DOI:** 10.3390/e23020201

**Published:** 2021-02-06

**Authors:** Qinfeng Xiao, Jing Wang, Youfang Lin, Wenbo Gongsa, Ganghui Hu, Menggang Li, Fang Wang

**Affiliations:** 1School of Computer and Information Technology, Beijing Jiaotong University, Beijing 100044, China; qfxiao@bjtu.edu.cn (Q.X.); yflin@bjtu.edu.cn (Y.L.); gongsa@bjtu.edu.cn (W.G.); ganghuihu@bjtu.edu.cn (G.H.); 2Beijing Key Laboratory of Traffic Data Analysis and Mining, Beijing 100044, China; 3Beijing Laboratory of National Economic Security Early-Warning Engineering, Beijing Jiaotong University, Beijing 100044, China; mgli1@bjtu.edu.cn (M.L.); wang.fang@bjtu.edu.cn (F.W.); 4National Academy of Economic Security, Beijing Jiaotong University, Beijing 100044, China

**Keywords:** anomaly detection, knowledge distillation, ensemble learning, deep learning, cross entropy, information entropy

## Abstract

We address the problem of unsupervised anomaly detection for multivariate data. Traditional machine learning based anomaly detection algorithms rely on specific assumptions of normal patterns and fail to model complex feature interactions and relations. Recently, existing deep learning based methods are promising for extracting representations from complex features. These methods train an auxiliary task, e.g., reconstruction and prediction, on normal samples. They further assume that anomalies fail to perform well on the auxiliary task since they are never trained during the model optimization. However, the assumption does not always hold in practice. Deep models may also perform the auxiliary task well on anomalous samples, leading to the failure detection of anomalies. To effectively detect anomalies for multivariate data, this paper introduces a teacher-student distillation based framework *Distillated Teacher-Student Network Ensemble* (DTSNE). The paradigm of the teacher-student distillation is able to deal with high-dimensional complex features. In addition, an ensemble of student networks provides a better capability to avoid generalizing the auxiliary task performance on anomalous samples. To validate the effectiveness of our model, we conduct extensive experiments on real-world datasets. Experimental results show superior performance of DTSNE over competing methods. Analysis and discussion towards the behavior of our model are also provided in the experiment section.

## 1. Introduction

Anomaly detection (a.k.a. outlier detection) [1,2] is referred to as detecting data points that significantly deviate from normal behaviors. Identifying anomalies for multivariate data always provides valuable information in various domains. For instance, anomalies in credit card transactions could imply online a fraud [3], while an unusual computer network traffic recording could signify unauthorized access [4]. Due to the great empirical value, efficient and accurate anomaly detection algorithms are desired.

Compared with normalities, anomalies are associated with unknownness, irregularity, and rarity [2]. Unknownness indicates that anomaly events can not be observed until they actually happen. Furthermore, irregularity means that the class structure of anomalies is highly heterogeneous. Last but not least, anomalies are rare in terms of collected data, leading to the problem of class imbalance. Due to the difficulty of collecting a large-scale labeled anomaly dataset, fully supervised methods are impractical in real-world scenarios. Unsupervised anomaly detection does not require labeled training data. However, they rely on assumptions on the distribution of data points. For example, several distance- and density-based approaches, e.g., Local Outlier Factor (LOF) [5], KNN (k-nearest neighbor) [6], and Isolation Forest (iForest) [7], assume that normalities reside in the high density area, whereas anomalies are far from the normal cluster. Probabilistic methods assume that normalities follow a specific distribution, e.g., ABOD (Angle-Based Outlier Detection) [8] and COPOD (Copula-Based Outlier Detection) [9]. Those methods fail to handle complex features because of high-dimensional features and sophisticated nonlinear relations.

Recently, a number of works have been introduced to improve anomaly detection by deep neural networks [2]. Deep learning [10] has had remarkable success in modeling intricate dependencies on a variety of domains. Instead of building assumptions of normality distribution, deep anomaly detection models the normal patterns by training networks via a surrogate task, such as reconstruction [11,12], prediction [13,14,15], and classification [16,17]. The underlying structure of normalities is captured through training a model on normal samples. It is intuitive that anomalies are expected to perform the surrogate task badly since they violate normal patterns learned by the model. However, existing deep anomaly detection methods are problematic for multivariate data. For instance, reconstruction-based methods are still ineffective since autoencoders tend to produce blurred reconstructions. Advanced generative models, e.g., generative adversarial nets [18], are promising to generate reliable reconstructions, whereas they suffer the difficulties of optimization. Prediction-based methods are predominantly designed for temporal data such as time-series and videos. Classification-based methods formulate the surrogate task as distinguishing the true type of a transformed image from different augmentations such as rotation, flipping, and cropping [16]. However, the augmentations require geometric structures contained in datasets, e.g., images and videos, which are impractical for multivariate data. Thus, a dedicated surrogate task is desired.

In this paper, we present a novel framework to achieve efficient and accurate anomaly detection. Novelties of this work are three fold. First, the framework of the teacher-student distillation is dedicatedly designed for multivariate data, enabling the capability to model complex feature interactions and relations. In addition, our proposed framework is general and can be extended to apply to other domains such as image anomaly detection and video anomaly detection. Second, the ensemble of student networks provides a capability to avoid potential generalizing of the auxiliary task on anomalous samples. Third, multiple anomaly scores are provided to detect anomalies from various aspects.

The paper is organized in the following order. Related works concerning state-of-the-art anomaly detection methods are reviewed in Section 2. Section 3 describes the overview of our proposed framework. In Section 4, a detailed description of the instantiated model of the framework is developed. The experimental setup, experimental results, and analysis are illustrated in Section 5. Finally, discussion and conclusions are presented in Section 6 and Section 7, respectively.

## 2. Related Work

There exists an abundance of works on unsupervised anomaly detection [2]. Traditional methods, such as LOF [5], SVDD (Support Vector Data Description) [19], and iForest [7], are ineffective at dealing with high-dimensional feature space or complex feature interactions. Although some works propose efficient techniques to deal with high-dimensional categorical feature’s interactions, e.g., CatBoost [20] and XGBoost [21], they are supervised methods and can not be directly applied on unsupervised anomaly detection. Deep anomaly detection (reviewed in [2]) becomes a vivid research area that includes various research topics such as utilizing prior knowledge [22,23] and representation learning for normalities [24,25]. However, unsupervised anomaly detection is still challenging due to the complexities of modeling nonlinear dependencies.

To tackle complex datasets, recent literature shows the trend of detecting anomalies via techniques including low-dimensional embedding, anomaly detector ensemble, and self-supervised pretext tasks.

### 2.1. Anomaly Detection via Low Dimensional Embedding

Performing anomaly detection on high-dimensional space is challenging since abnormal characteristics of anomalies are always invisible in an original space. A widely used solution is to reduce the dimension of original features where anomalies become noticeable in a reduced low-dimensional space. Traditional methods focusing on identifying anomalies in a subset of original features or a constructed subspace. These approaches can be categorized into two sub-categories, including subspace-based methods [26,27] and feature-selection based methods [28,29,30]. However, discovering complex feature interactions and couplings is still challenging for traditional algorithms. Recently, several methods leverage the power of deep learning and representation learning. REPEN, an instance of the framework RAMODO proposed in [24], combining with anomaly detection and representation learning, learns customized low-dimensional embeddings of ultrahigh-dimensional data for a random distance-based outlier detector. RDP [25] is a generic representation learning method by predicting data distances in a random projection space. By incorporating dedicated objectives, the low-dimensional representations obtained by RDP can be further used to perform various downstream tasks such as anomaly detection and clustering. Despite utilizing deep neural networks and representation learning, these methods learn representations and detect anomalies separately, which lead to indirect optimization and unstable performance.

### 2.2. Anomaly Detector Ensemble

Ensembling weak anomaly detectors into a strong anomaly detector is a powerful technique to achieve accurate anomaly detection. Traditional methods such as Feature Bagging [27], iForest [7], and LODA [31] try to combine outcomes from weak anomaly detectors to produce an ensembled anomaly score. However, these methods based on simple anomaly detectors and thus fail to model high-dimensional complex features.

### 2.3. Anomaly Detection with Self-Supervised Pretext Tasks

Recent self-supervised learning algorithms draw support from a pretext task to learn semantic representations. Different from representation learning, several methods utilize self-supervised pretext tasks to detect anomalies. GEOM [16] builds up a framework of self-supervised learning for anomaly detection. Specifically, GEOM applies multiple geometric transformations to an instance and trains a classifier to correctly predict the transformation giving a transformed instance. $E^{3}$-Outlier [17] follows the idea of GEOM, adding up multiple anomaly scores. GOAD [32] substitutes the classification task in GEOM with a one-class classification task. However, the geometric transformations used in these methods require geometric structures contained in the dataset, which can not be directly applied to multivariate datasets.

## 3. The Distillation-Based Anomaly Detection Framework

### 3.1. Problem Statement

We address the problem of fully unsupervised anomaly detection for multivariate data. In detail, the framework consists of a triplet 〈M,T,H〉, where M, T, and H stand for the model, the surrogate task, and the anomaly scoring function, respectively. By training the model M via the surrogate task T unsupervised and building the anomaly scoring function H upon T, the goal of anomaly detection is to assign anomaly scores to test samples in a way that we have $H\left(x_{i}\right)>H\left(x_{j}\right)$, where $x_{i}$ is an anomaly and $x_{j}$ is a normal object.

Additionally, whether a test sample x is an anomaly is judged by specifying an anomaly threshold λ:(1)$$A\left(x\right)=\left\{\begin{array}{cc} 1, & \mathrm{if}\;H(x)>\lambda \\ 0, & \mathrm{Otherwise} \end{array}\right.,$$
where A(x) is the test result of x.

### 3.2. The Proposed Framework

To tackle the problem of unsupervised anomaly detection, we introduce a novel framework including a teacher-student network and corresponding anomaly scores. In particular, the teacher network learns low-dimensional embeddings on the whole feature space, whereas only normal patterns are distillated from the teacher to student networks through a regression loss. Anomalies are highlighted by anomaly scores since gaps between the teacher network and student networks are expected to be large and uncertain. As shown in Figure 1, the framework contains three major modules:**Teacher network** is a parametric function T(·) that maps original features to low-dimensional vectors. We require the teacher network to have two properties. First, the teacher network is able to preserve the distance information of original space. Namely, two instances that are close in the original space are still close in the low-dimensional space. Second, the teacher network is a smooth and injective function such that the normality manifold is formed in the low-dimensional space, whereas anomalies are laid outside of the manifold.**Student networks** are a group of parametric functions ${\left\{S^{\left(k\right)}\left(\cdot \right)\right\}}_{k=1\cdots K}$. They are trained to mimic the outputs of the teacher network only on normal samples. All the students are independently trained to improve robustness. Anomalies are detected when the students fail to generalize the teacher mapping outside the normality manifold.**Anomaly scores**. By investigating the gaps between the teacher network and the student networks, we can define an anomaly score H(·) to identify anomalies. The anomalous samples are expected to gain larger gaps since only knowledge of normal patterns are distillated. In addition, the variance of students can be used as an additional criterion to detect anomalies.

To instantiate each component of the framework, we need to answer following questions:**Q1**: How can design the structure and the training objective of the teacher network be designed?**Q2**: How can the student networks be trained such that they mimic behaviors of the teacher only on normal samples?**Q3**: How can an anomaly score be defined that can effectively identify anomalous objects?

These questions are answered in Section 4.1, Section 4.2 and Section 4.3, respectively. Thus, we present an instantiated method of the framework called *Distillated Teacher-Student Network Ensemble* (DTSNE).

## 4. Distillated Teacher-Student Network Ensemble

### 4.1. Teacher Network for Low-Dimensional Embedding

Let $R^{L}$ be the original feature space where *L* is the dimension of $R^{L}$. Normal samples lie in a subspace $X\subset R^{L}$, whereas anomalous samples lie in a complementary space $X^{\prime }=R^{L}\backslash X$ outside of the normal space X. The teacher network $T_{\theta }\left(\cdot \right):R^{L}\mapsto R^{M}$ that maps each instance in $R^{L}$ to a low-dimensional space $R^{M}$ (*M* is the dimensional of the low-dimensional space) is parameterized by weights θ. We implement the teacher network by a multi-layer neural network with ReLU [33] activation functions. Assuming that $T_{\theta }\left(\cdot \right)$ is a smooth injective mapping, low-dimensional representations $z=T_{\theta }\left(x\right)$ of normal instances x∈X are expected to form a subspace $Z\subset R^{M}$ where anomalies also lie in a complementary space $Z^{\prime }=R^{M}\backslash Z$. As described in Section 3.2, we require that the teacher network be able to map the normal instance to a normality manifold Z and meanwhile map the anomalous outside Z. To achieve this, the most direct way is to gather normal instances into a cluster and pull anomalous instances out when labels are available. However, this approach is infeasible under an unsupervised setting. Thus, we propose a self-supervised objective to discover the intrinsic regularities.

#### Pre-Training with Self-Supervised Pretext Tasks

Self-supervised methods learn semantic representations by applying a surrogate task such as context prediction [34], solving jigsaw puzzles [35], distinguishing image rotations [36], etc. In this paper, we present the learning objective as distinguishing instance augmentations. Like [16], multiple transformations are applied on each instance. The teacher network plus a classification layer is trained to predict the correct transformation of a given instance. The procedure is illustrated in Figure 2. In detail, we assume that the training dataset X only contains normal samples (the problem of anomaly contamination is discussed in Section 4.3). We further use a group of affine transformations to transform the dataset:(2)$$T\left(x;k\right)=\left\{\begin{array}{cc} W_{k}x+b_{k}, & \mathrm{if}\;k\neq 0\\ Ix, & \mathrm{if}\;k=0, \end{array}\right.$$
where $W_{k}$ is the random weight matrix and $b_{k}$ is the random affine vector. Specifically, T(x;k) is an identity mapping when k=0. Thus, the augmented dataset $X_{T}$ is given by:(3)$$X_{T}≜\left\{T\left(x;k\right):x\in X,k=0\cdots K\right\},$$
where x is an instance of the training dataset X, and *K* is the number of transformations.

Conventional self-supervised learning always considers transformed instances belonging to the same semantic class. In contrast, we treat transformed instances as “pseudo-anomalies”, i.e., they do not belong to the normality manifold X. Lastly, we design the training objective as distinguishing transformed instances. The probability that an instance is correctly classified is given by:(4)$$P\left(y=k|T\left(x\right);f_{\eta }\left(\cdot \right),k\right)=\mathrm{Softmax}\left(f_{\eta }\left(T\left(x;k\right)\right)\right),$$
where Softmax(·) is the Softmax function, and $f_{\eta }\left(\cdot \right)$ is a classification layer.

The final training objective of the teacher network is:(5)$$L_{T}=-\sum_{x\in X}\sum_{k}\log P\left(y=k|T\left(x\right);f_{\eta }\left(\cdot \right),k\right).$$

The pseudo-code of the teacher network pre-training is illustrated in Algorithm 1.

By distinguishing transformed instances, the representations that the teacher network produces are optimized to gather normal representations z∈Z, whereas pull anomalous representations $z\in Z^{\prime }$ automatically in an unsupervised manner. In addition, we found that it also brings benefits to anomaly scoring. The details are discussed in Section 4.3:
**Algorithm 1** Teacher Network Pre-trainingINPUT: Teacher network $T_{\theta }$, training dataset X, the number of transformation *K*
OUTPUT: Pre-trained teacher network $T_{\theta }$
 1:Initialize the teacher network $T_{\theta }\left(\cdot \right)$ 2:Initialize the dataset $X_{T}$ 3:**for**i=0,1,⋯K**do** 4:  Calculate the *k*-th transformed dataset $X_{k}$ using Equation (2) 5:  Add $X_{k}$ to $X_{T}$ 6:**end for** 7:Optimize the loss function described in Equation (5)

A simplified version of the teacher network is just using **Random Initialization** without training. We show that Random Initialization is still capable of capturing internal regularities in the experimental section. Dedicated comparison of the two pre-training paradigms is conducted in Section 5.6. Kaiming Initialization [37] is used for the initialization strategy.

### 4.2. Distillated Students for Normality Learning

Now, we describe how to train student networks $S_{\psi }^{\left(n\right)}\left(\cdot \right)$ (n∈{1,⋯,N}, where *N* is the number of students) by utilizing the supervisory of the teacher network. All students are implemented by the same structure, a multi-layer neural network, and trained independently. The training objective contains two criterions. The first criterion is to optimize the instance-wise distance directly:(6)$$L_{\mathrm{Inst}}\left(x\right)={\left\|T_{\theta }\left(x\right)-S_{\psi }^{\left(n\right)}\left(x\right)\right\|}_{2}.$$

The second criterion is the pairwise distance that encapsulates pairwise similarity information:(7)$$L_{\mathrm{Pair}}\left(x_{i},x_{j}\right)={\left\|l\langle T_{\theta }\left(x_{i}\right),T_{\theta }\left(x_{j}\right)\rangle -l\langle S_{\psi }^{\left(n\right)}\left(x_{i}\right),S_{\psi }^{\left(n\right)}\left(x_{j}\right)\rangle \right\|}_{2}.$$

For computation efficiency, the distance l〈·,·〉 is measured by cosine similarity:(8)$$L_{\mathrm{Pair}}\left(x_{i},x_{j}\right)={\left\|T_{\theta }\left(x_{i}\right)\cdot T_{\theta }\left(x_{j}\right)-S_{\psi }^{\left(n\right)}\left(x_{i}\right)\cdot S_{\psi }^{\left(n\right)}\left(x_{j}\right)\right\|}_{2}.$$

The final objective of student network training is given as:(9)$$L_{S}=L_{\mathrm{Inst}}+\alpha L_{\mathrm{Pair}},$$
where α is a hyperparameter to adjust the weight of pairwise distance.

By only training on normal samples, the students manage to accurately regress the features solely for normal samples. They yield large regression errors and predictive uncertainties for anomalous samples.

### 4.3. Anomaly Scores for Anomaly Identification

In the evaluation phase, anomalous test samples are expected to receive larger gaps between the teacher network and student networks. The anomaly score measures how anomalous an instance is. In DTSNE, anomalies are identified by the anomaly score from two perspectives: closeness and uncertainties. Giving a predefined criterion $H_{*}\left(x;n\right)$, the closeness and the uncertainties are obtained by calculating the mean value and the variance over each students, respectively:(10)$$H=E_{n}\left\{H_{*}\left(x;n\right)\right\}+\beta {\mathrm{Var}}_{n}\left\{H_{*}\left(x;n\right)\right\},$$
where β is the weight of student variance.

To instantiate Equation (10), various dedicated anomaly scores $H_{*}\left(x;n\right)$ can be defined. Specifically, a simple choice is the Mean Square Error (MSE):(11)$$H_{\mathrm{MSE}}\left(x;n\right)={\left\|T_{\theta }\left(x\right)-S_{\psi }^{\left(n\right)}\left(x\right)\right\|}_{2}.$$

In addition, advanced anomaly scores can be defined to utilize the information of self-supervised pre-training.


**Classification Probability**
Since the teacher network is trained to distinguish different instance transformations, students are expected to perform the classification task well on normal samples. One can directly use the Softmax response as the anomaly score:
(12)$$H_{\mathrm{CP}}\left(x;n\right)=\sum_{k}^{K}P\left(y=k|T\left(x\right);f_{\eta }\left(\cdot \right),k\right).$$This anomaly score measures the correctness of the student prediction giving each transformation of the instance x.
**Cross Entropy**
For each instance, we get *K* transformations in total. In addition, the *k*-th element of the Softmax response indicates the probability that the correct transformation index is *k* in terms of a transformed instance. Classification Probability (Equation (12)) only uses the *k*-th element of the Softmax response for the *k*-th transformation. To utilize information of all elements, we define the Cross Entropy based anomaly score:
(13)$$H_{\mathrm{CE}}\left(x;n\right)=-\sum_{k}^{K}y\log P\left(y=k|T\left(x\right);f_{\eta }\left(\cdot \right),k\right).$$Here, the Cross Entropy measures the closeness of the Softmax response and the ground truth distribution.
**Information Entropy**
Since student networks are trained to mimic the teacher network only on normal samples, we claim that the Softmax response of an anomalous sample is more chaotic than a normal sample’s. We can use Information Entropy to measure the prediction confidence:
(14)$$H_{\mathrm{IE}}\left(x;n\right)=-\sum_{k}^{K}\log P\left(y=k|T\left(x\right);f_{\eta }\left(\cdot \right),k\right).$$

In this paper, Mean Square Error and Information Entropy are used for anomaly scoring. Thus, the final anomaly score is given:(15)$$H_{\mathrm{Final}}\left(x;n\right)=H_{\mathrm{MSE}}\left(x;n\right)+\gamma H_{\mathrm{IE}}\left(x;n\right),$$
where γ is the weight of Information Entropy score.

#### Dealing with Anomaly Contamination

In Section 4.1, we assume that all the samples in the training dataset X are normal, i.e., they are sampled from the normality manifold X. However, this assumption is not guaranteed under an unsupervised scenario. This problem is called Anomaly Contamination. To alleviate this issue, we drop a portion of samples that have the highest anomaly scores per iteration to iteratively improve the pureness of the training dataset. This technique is also used in [25].

### 4.4. Summary

In summary, we propose the instantiation of our framework: DTSNE. The workflow contains three steps:**Teacher network pre-training.** First, the teacher network is first pre-trained with a self-supervised objective (Equation (5)) to learn underlying regularities of the data. In detail, we transform each instance using multiple random transformations and train the teacher network (plus a linear classification layer) to distinguish them. This objective helps the teacher network to produce better embeddings for downstream anomaly detection.**Training of student networks.** Second, the parameters of the teacher network are frozen. The teacher network can be treated as a function that provides supervision for students. Multiple student works are trained to mimic the outcomes of the teacher using the objective of Equation (9) only on normal samples.**Anomaly scoring.** Lastly, anomalies are identified by calculating an anomaly score. It evaluates the outcomes of students from a specific perspective, such as the difference between the teacher. In addition, we also use the uncertainties among students since they tend to produce contradictory outcomes for anomalous instances.

## 5. Experimental Results

### 5.1. Datasets

As shown in Table 1, the experiments are conducted on 10 publicly available datasets from various domains, including network intrusion detection, fraud detection, medical disease detection, etc. Two datasets contain real anomalies, including Lung and U2R, while other datasets are transformed from extremely imbalanced datasets. Following [22,25,38,39], the rare class in the imbalanced dataset is treated as semantically anomalies. Categorical features are encoded into discrete values via one-hot encoding. Lastly, all features are rescaled to the range [0,1] using a min-max scaler to ensure the optimization efficiency.

### 5.2. Competing Methods

We consider various anomaly detection methods for evaluation, including traditional methods, e.g., LOF, outlier ensemble methods, e.g., iForest and LODA (*Lightweight Online Detector of Anomalies*), and state-of-the-art deep learning methods, e.g., DAGMM (*Deep Autoencoding Gaussian Mixture Model*) and RDP (*Random Distance Prediction*). The short descriptions of those methods are summarized as follows:**iForest** [7,40] is an outlier ensemble method that detects anomalies by selecting a random feature and then splits instances by a randomly selected splitting point into two subsets. The partitioning is applied recursively until a predefined termination condition is satisfied. Since the recursive partitioning can be represented by a binary tree structure, it is expected that anomalies have noticeably shorter paths started from the root node.**LOF** [5] is a density-based anomaly detection algorithm that detects anomalies by measuring the local deviation of a given instance w.r.t. its neighbors. The local deviation of an instance is given by the local density, which is calculated by the ratio of the KNN distance of the object to its k-nearest neighbors’ KNN distances. Under the assumption that anomalies lie in the low-density area, the local density of an anomaly is expected to be substantially lower than normal objects.**LODA** [31] is a lightweight ensemble system for anomaly detection. LODA groups weak anomaly detectors into a strong anomaly detector, which is robust to missing variables and identifying causes of anomalies.**DAGMM** [41] is composed of two modules, a deep autoencoder and a Gaussian Mixture Model (GMM). Instead of training the two components sequentially, DAGMM jointly optimizes the parameters of the two modules in an end-to-end fashion. This training strategy balances autoencoding reconstruction and density estimation of latent representations well, achieving a better capability of stable optimization and thus further reducing reconstruction errors.**RDP** [25] first trains a feature learner to predict data distances in a randomly projected space. The training process is flexible to incorporate auxiliary loss functions dedicatedly designed for downstream tasks such as anomaly detection and clustering. The representation learner is optimized to discover the intrinsic regularities of class structures that are implicitly embedded in the randomly projected space. Lastly, anomalies can be identified by calculating the distance between the representation learner and the random projection.

### 5.3. Implementation Details

This section provides the detailed information of our implementation. Since the evaluation focuses on multivariate data, multi-layer perceptrons (MLP) are used. Specifically, both the teacher network and the student network consist of two perceptron layers and a LeakyReLU activation function. In pre-training, the transformation matrices are sampled from Gaussian distribution. To perform classification, an additional classification layer is used, which is composed of a simple perceptron layer. In the training phase, the parameters of the teacher network are frozen. We use the SGD (Stochastic Gradient Descent) optimizer with momentum and L2 penalty. The learning rate, the momentum factor, and the L2 penalty factor are set as 0.1, 0.9, and 0.001, respectively. We apply gradient clipping to avoid overfitting, with a coefficient of 0.1. If self-supervised pre-training is enabled, we pre-train the teacher network using 100 epochs across all datasets. Furthermore, the student networks also trained using 100 epochs for each dataset.

### 5.4. Evaluation Metrics

Following related works [22,23,24,25], we focus on evaluating the score function H instead of the test result A (see Equation (1)). To achieve this, we use the area under the Receiver Operating Characteristic (ROC) curve, denoted as AUROC, and the mean Average Precision (mAP) as the performance metrics. In addition, all experiments are conducted at a virtual container with 32 GB dedicated memory, four Xeon CPU cores and four Geforce GTX 2080 GPU.

### 5.5. Parameter Settings

We choose the two factor α (the weight of pairwise distance, in Equation (9)) and β (the weight of student variance, in Equation (10)) empirically. Through quantitative experiments, we found the influence of α and β is very marginal when they are relatively small (see Section 5.7). To avoid over-parameterization, the two parameters are both set as 1 for simplification. Due to the two parameters: the number of students and the number of transformations significantly influence the computational efficiency, we empirically set them as 8 and 32, respectively. The portion of samples to be dropped is set as 5%. We perform eight dropping iterations for the Apascal dataset and four iterations for other datasets. Other parameter settings such as batch size, embedding dimension, and γ are selected based on the dataset. Table 2 introduces the settings of them for each dataset respectively.

### 5.6. Performance on Real-World Datasets

**Experiment settings.** This section compares the performance of DTSNE versus competing methods on real-world datasets. Each data set is first randomly split into two subsets, with 70% instances as the training set and the other 30% instances as the evaluation set. To ensure a reliable evaluation, 10 trials are independently performed with different random seeds. The performance is measured by AUROC and mAP scores. The implementations of iForest and LOF are taken from scikit-learn, a widely used machine learning library. All parameters of the two methods are set as default values. We implement LODA by APIs offered by PyOD [42], a comprehensive and scalable Python toolkit for anomaly detection. The parameters keep the default values from PyOD. DAGMM and RDP are implemented by PyTorch [43]. All the implementations of DAGMM and RDP and parameter settings are adopted from the code released with the original papers.

**Analysis.** The teacher-student distillation based framework enables DTSNE to achieve superior performance over competing methods. Table 3 and Table 4 report the AUROC scores and mAP scores, respectively (highest score highlighted in bold). In terms of AUROC scores, DTSNE achieves substantially better average performance than iForest (130.4%), LOF (77.2%), LODA (34.6%), DAGMM (30.4%), and RDP (7.2%); The improvements of our proposed method w.r.t. mAP scores is much more significant than iForest (561.7%), LOF (466.1%), LODA (111.8%), DAGMM (86.3%), and RDP (10.5%). These results show that our method can effectively deal with complex multivariate data. In detail, DTSNE outperforms LOF on all datasets w.r.t. both AUROC and mAP. This proves that simple density-based methods are ineffective in tackling complex feature interactions and relations. In addition, our proposed method also outperforms the two outlier-ensemble based methods, iForest and LODA, on all datasets. Compared with simply stacking weak anomaly detectors, our method utilizes multiple student networks and identifies anomalies from both the outcomes of student networks and disagreements among students. Lastly, DTSNE beats competing deep anomaly detection methods DAGMM and RDP as well. As for DAGMM, our method achieves better AUROC scores over all datasets and better mAP scores on almost all datasets except Secom. It has been proven that reconstruction-based anomaly detection is ineffective at dealing with multivariate data though deep neural networks that are used. In terms of RDP, DTSNE obtains better performance on almost all datasets. Instead of only investigating the gaps between the teacher and students, our method incorporates customized anomaly scores to leverage additional information contained in the outcomes of students. Except for that, the ensemble of student networks also contributes the improvements over RDP.

**Significance test.** To investigate the significance of our proposed method, we perform a Friedman test [44] w.r.t. AUROC and mAP scores, respectively. A Friedman test is a non-parametric equivalent ANOVA (Analysis of variance) for multiple datasets. We further visualize the significance result in Figure 3. Two non-overlapping segments indicate “statistically difference” between the two algorithms. The test results suggest a significant difference between our method and baselines except RDP because a small overlap is observed. However, RDP is a state-of-the-art algorithm and the performance is already very high. We can still infer that DTSNE is much better than RDP since our method outperforms it on almost all datasets and the overlapping is extremely small.

### 5.7. Sensitivity Test w.r.t. Hyper-Parameters

**Experiment settings.** This section conducts the sensitivity test w.r.t. three perspectives: the representation dimension, the parameter α, and the anomaly score. To empirically investigate the influence of them, we conduct quantitative experiments w.r.t. different settings on all datasets. The candidate values of the representation dimension are 4, 8, 16, 32, 64, 128, and 256, while the settings of α include 0.1, 0.2, 0.5, 1, 2, 5, and 10.

**Impact of representation dimensions.**Figure 4 reports the AUROC scores and mAP scores of DTSNE w.r.t. different representation dimensions, respectively. The experimental results demonstrate the stability of our proposed method on different parameter settings across datasets. In terms of AUROC scores, the experimental results suggest that extremely large representation dimensions can not provide better performance. This illustrates that mapping original features into a high-dimensional representation space is not helpful for anomaly detection since anomalous patterns tend to be invisible in a higher-dimensional space. The experimental results also show extremely small representation dimensions can not provide sufficient information for the teacher network to learn a semantically rich representation. In terms of mAP scores, four datasets (AID362, Chess, CMC, and Secom) show flat trends in Figure 4. This suggest a good performance can be achieved with fairly small representation dimensions on the four datasets. Other datasets achieve best performance at a “elbow ” point. The settings of representation dimensions can be empirically selected as the “elbow” points to guarantee an accurate anomaly detection.

**Impact of α.**Figure 5 illustrates the influence of α. The performance of DTSNE is relatively stable w.r.t. different α values. When α is extremely large, a rapid drop can be investigated for AD and R10 in terms of mAP scores. This phenomenon suggests to us to choose a relatively small value for α. To avoid over-parameterization, we set α as 1 for all datasets.

**Impact of anomaly scores.**Figure 6 and Figure 7 show the AUROC scores and mAP scores in terms of different anomaly scores separately. The experimental results suggest that none of the anomaly scores suppress others. Although Information Entropy and Classification Probability obtain improvement only in some cases, it never harms the performance. Cross Entropy performs worse on dataset AID362, Bank, and Lung, but it also achieves improvements on dataset Apascal, CMC, and Secom. To summarize, the choice of anomaly scores heavily relies on characteristics of datasets. In this paper, we use Information Entropy as the additional anomaly score since it performs stably on all datasets.

### 5.8. Ablation Study

**Experiment settings.** In this section, we examine the effectiveness of the key component of DTSNE, self-supervised pre-training, by comparing the performance of random initialization and self-supervised pre-training. The variant is called ${\mathrm{DTSNE}}_{\mathrm{random}}$, where the self-supervised pre-training is disabled. The teacher network of ${\mathrm{DTSNE}}_{\mathrm{random}}$ is randomly initialized without training. The network structure and training settings of DTSNE and its variants are totally the same.

**Analysis.** The experimental results show the importance of applying self-supervised pre-training for the teacher network. Table 5 shows the performance of DTSNE and its variant. Specifically, DTSNE boosts up AUROC scores (20.5%) and mAP scores (96.4%), respectively, than variants. As discussed in Section 4.1, the reason for such improvement is that self-supervised pre-training enables our model to preserve intrinsic regularities of the normal manifold, which makes anomalies more noticeable in the representation space simultaneously. The experimental results also show that the technique is widely applicable on various datasets since they are collected from different domains with different dimensions (ranging from tens to thousands). Moreover, it is also noticeable that the variant ${\mathrm{DTSNE}}_{\mathrm{random}}$ obtains favorable results on most of the datasets. This phenomenon suggests that a random smooth function to some extent has the ability to reflect interactions from the original space. More discussion can be found in [25].

## 6. Discussion

**Anomaly Detection in Low-dimensional Embedding.** Detecting anomalies in a reduced space is a widely used solution to tackle high-dimensional complex feature space. In this paper, the reasons for learning low-dimensional representations are two fold: first, in contrast to operating on original space, detecting anomalies on low-dimensional space. Second, one can assume that normal data points are lying inside the normality manifold, whereas anomalies are lying outside. By mapping data points into a lower dimensional compact space, regularities are preserved while irregularities are highlighted.

**Self-supervised Pre-training.** Compared with using a random projection like [25], we pre-train the teacher network with a self-supervised objective. Though a random projection implemented by a neural network has some good properties with such smoothness, the ability to preserve the normal manifold can not be guaranteed without any regularization, which may potentially map anomalies into an area close to the normal cluster. As a result, the self-supervised pre-training is thus an essential component in DTSNE. We believe the importance of using self-supervised pre-training is also true in other anomaly detection algorithms.

**Ensemble of Weak Anomaly Detectors.** Conventional ensemble anomaly detection aims at effectively aggregating detection results from weak anomaly detectors. Different from that, our ensemble approach considers an additional perspective: uncertainties of weak anomaly detectors (namely students). Diversities among students are guaranteed by different initialization states. Since knowledge about anomalies is not accessible during the training phase, the uncertainties among weak anomaly detectors is an important sign to identify anomalous instances. This technique is readily applicable to other ensemble-based anomaly detection algorithms.

## 7. Conclusions

In this paper, we proposed a teacher-student distillation based anomaly detection framework and a corresponding instantiation DTSNE (*Distillated Teacher-Student Network Ensemble*). Key components including self-supervised pre-training, teacher-student distillation, student ensemble, and multiple anomaly scores enable DTSNE to greatly outperform state-of-the-art anomaly detection algorithms. The effectiveness of our method to detect anomalies for multivariate data is justified by extensive experiments. DTSNE achieves a substantially better average performance 7.2% and 10.5% than RDP (a state-of-the-art anomaly detection method) in terms of AUROC scores and mAP scores, respectively. The experimental results imply broader impacts of our techniques: self-supervised pre-training and student ensemble.

## 8. Future Work

The proposed approach achieves promising performance for unsupervised anomaly detection on various real-world datasets. The success suggests to us to explore future works on refining the current framework. One of the drawbacks of our approach is that the improvements of additional anomaly scores are not significant. The reason is that the information provided by outcomes of students is not fully utilized. To make the best use of information of the outcomes, we propose that the future work could look at developing a criteria to measure the uncertainties of student networks instead of only using variance. Similar works in recent literature, such as Bayesian Neural Networks [45] and Predictive Uncertainty [46], can provide inspiration to achieve the goal.

Furthermore, the future work could be explored on incorporating partial available labels. It is natural to occasionally obtain some annotations in real-world applications. As recent literature [23,47] suggests, only partial labels can significantly improve the performance of anomaly detection compared to a fully unsupervised scenario. Therefore, future work needs to extend our framework to adopt the semi-supervised setting.

## Figures and Tables

**Figure 1 entropy-23-00201-f001:**
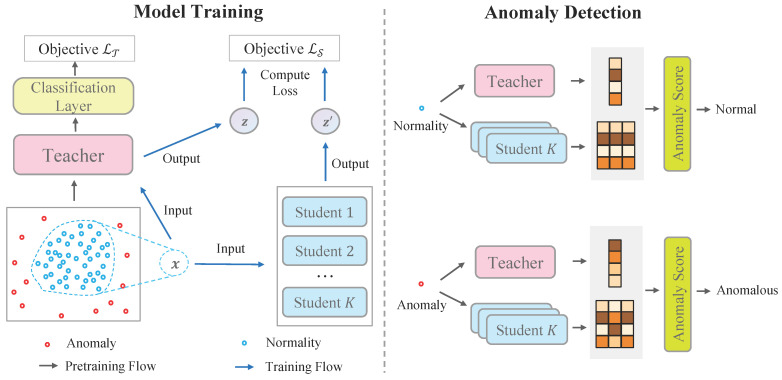
Overview of our framework. The left-hand side illustrates the training procedure of the teacher network and student networks. The right-hand side shows the anomaly detection procedure.

**Figure 2 entropy-23-00201-f002:**
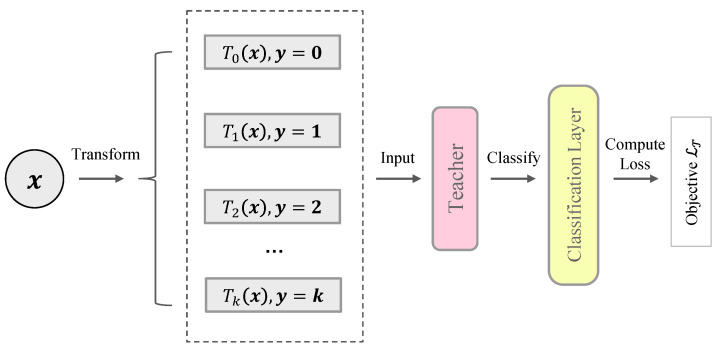
Illustration of the teacher network pre-training. The pre-training means that we train the teacher network using a predefined objective before the main procedure of training student networks.

**Figure 3 entropy-23-00201-f003:**
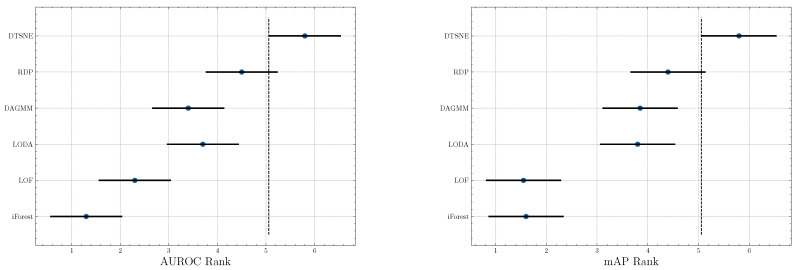
Graphical presentation of the Friedman significance test w.r.t. AUROC and mAP scores.

**Figure 4 entropy-23-00201-f004:**
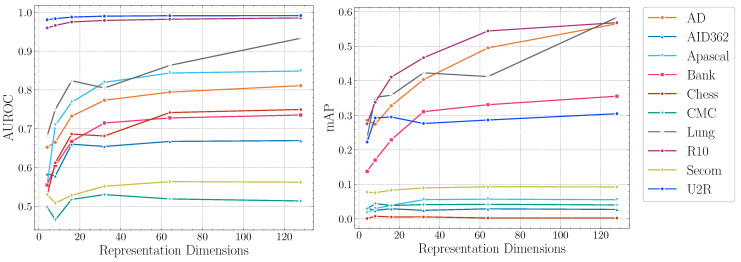
AUROC and mAP scores of DTSNE w.r.t. representation dimensions.

**Figure 5 entropy-23-00201-f005:**
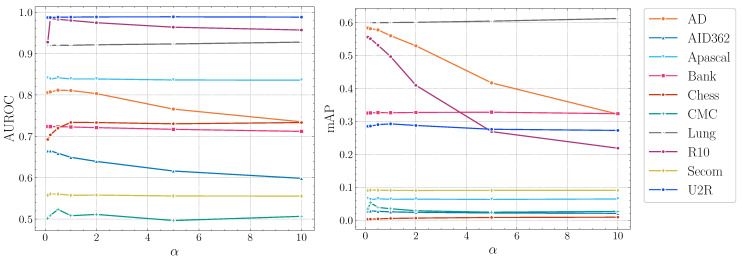
AUROC and mAP scores of DTSNE w.r.t. different α settings.

**Figure 6 entropy-23-00201-f006:**
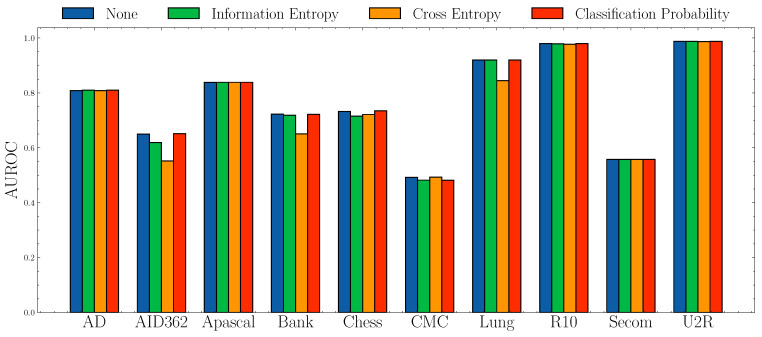
AUROC socres w.r.t. different anomaly scores.

**Figure 7 entropy-23-00201-f007:**
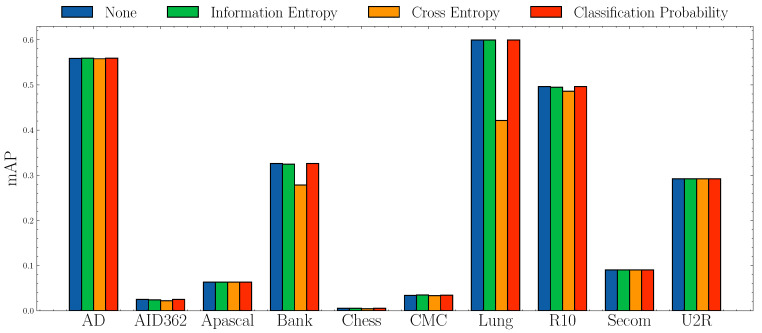
mAP socres w.r.t. different anomaly scores.

**Table 1 entropy-23-00201-t001:** Statistics of datasets.

Dataset	# Instances	Dimension	Anomaly Ratio
AD	3279	1558	14.00%
AID362	4279	114	1.40%
Apascal	12,695	64	1.39%
Bank	41,188	62	11.26%
Chess	28,056	23	0.10%
CMC	1473	8	1.97%
Lung	145	3312	4.14%
R10	12,897	100	1.84%
Secom	1567	590	6.64%
U2R	60,821	34	0.37%

**Table 2 entropy-23-00201-t002:** Parameter settings of different datasets. γ is the factor of the additional anomaly score (see Equation (15)).

Dataset	Batch Size	Embedding	γ
AD	128	128	0.0001
AID362	256	32	0.01
Apascal	256	32	0.00001
Bank	256	50	0.01
Chess	1024	16	0.01
CMC	32	4	0.01
Lung	128	128	0.01
R10	128	32	0.01
Secom	32	64	0.0001
U2R	1024	16	0.001

**Table 3 entropy-23-00201-t003:** AUROC scores of DTSNE and baselines on 10 datasets. The best performance for each dataset is boldfaced.

Dataset	iForest	LOF	LODA	DAGMM	RDP	DTSNE
AD	0.3959±0.0366	0.5343±0.0118	0.5280±0.0996	0.5033±0.0289	0.8164±0.0001	0.8090±0.0175
AID362	0.4246±0.0303	0.4896±0.0415	0.5491±0.1023	0.5079±0.0194	0.5952±0.0208	0.6804±0.0285
Apascal	0.5059±0.0264	0.5079±0.0175	0.5714±0.1813	0.5438±0.0973	0.8010±0.0031	0.8367±0.0303
Bank	0.3846±0.0079	0.5190±0.0024	0.5114±0.0446	0.5595±0.0216	0.7503±0.0003	0.7408±0.0054
Chess	0.4902±0.0466	0.3829±0.0738	0.5145±0.1618	0.4984±0.0003	0.4824±0.1216	0.7443±0.1576
CMC	0.4456±0.0651	0.5097±0.0314	0.5300±0.1070	0.5151±0.0173	0.4279±0.0876	0.5304±0.0986
Lung	0.1843±0.1575	0.4556±0.1408	0.8552±0.1116	0.6869±0.1202	0.9210±0.0007	0.9332±0.0921
R10	0.0550±0.0061	0.0651±0.0090	0.4961±0.2039	0.7874±0.0361	0.9845±0.0024	0.9855±0.0016
Secom	0.4542±0.0349	0.4728±0.0271	0.5259±0.0627	0.5203±0.0203	0.5590±0.0016	0.5596±0.0656
U2R	0.0481±0.0010	0.4692±0.0177	0.7201±0.2493	0.8650±0.0667	0.9484±0.0010	0.9881±0.0027
Average	0.3388±0.0413	0.4406±0.0373	0.5802±0.1324	0.5987±0.0428	0.7286±0.0239	0.7808±0.0500

**Table 4 entropy-23-00201-t004:** mAP scores of DTSNE and baselines on 10 datasets. The best performance for each dataset is boldfaced.

Dataset	iForest	LOF	LODA	DAGMM	RDP	DTSNE
AD	0.1196±0.0117	0.1448±0.0115	0.2124±0.0628	0.1480±0.0258	0.5088±0.0014	0.5598±0.0385
AID362	0.0145±0.0020	0.0130±0.0023	0.0224±0.0088	0.0204±0.0093	0.0237±0.0041	0.0287±0.0044
Apascal	0.0129±0.0017	0.0140±0.0011	0.0245±0.0237	0.0575±0.0215	0.0419±0.0001	0.0599±0.0094
Bank	0.0953±0.0012	0.1170±0.0015	0.1231±0.0209	0.3326±0.0152	0.3314±0.0007	0.3642±0.0152
Chess	0.0010±0.0004	0.0009±0.0002	0.0012±0.0004	0.0009±0.0002	0.0013±0.0005	0.0088±0.0152
CMC	0.0170±0.0058	0.0230±0.0055	0.0256±0.0117	0.0381±0.0123	0.0246±0.0056	0.0413±0.0238
Lung	0.0450±0.0231	0.0631±0.0244	0.4346±0.2587	0.2487±0.1092	0.7178±0.0057	0.5838±0.2572
R10	0.0172±0.0019	0.0166±0.0017	0.1806±0.1392	0.2992±0.0384	0.5311±0.0459	0.5684±0.0511
Secom	0.0677±0.0058	0.0645±0.0106	0.0851±0.0258	0.1999±0.0082	0.0834±0.0001	0.0932±0.0184
U2R	0.0037±0.0003	0.0035±0.0003	0.1212±0.0993	0.0535±0.0103	0.0951±0.0029	0.2982±0.1006
Average	0.0394±0.0054	0.0460±0.0059	0.1231±0.065	0.1399±0.0251	0.2359±0.0067	0.2606±0.0534

**Table 5 entropy-23-00201-t005:** AUROC scores and mAP scores of DTSNE variants.

Dataset	${\mathrm{DTSNE}}_{\mathrm{random}}$		DTSNE	
	AUROC	mAP	AUROC	mAP
AD	0.6810±0.0541	0.2513±0.0543	0.8090±0.0175	0.5598±0.0385
AID362	0.6544±0.0487	0.0243±0.0050	0.6804±0.0285	0.0287±0.0044
Apascal	0.7772±0.0405	0.0350±0.0120	0.8367±0.0303	0.0599±0.0094
Bank	0.7209±0.0036	0.3071±0.0209	0.7408±0.0054	0.3642±0.0152
Chess	0.6699±0.1438	0.0153±0.0368	0.7443±0.1576	0.0088±0.0152
CMC	0.4963±0.1305	0.0312±0.0182	0.5304±0.0986	0.0413±0.0238
Lung	0.8569±0.1196	0.3796±0.2463	0.9332±0.0921	0.5838±0.2572
R10	0.0889±0.0216	0.0101±0.0009	0.9855±0.0016	0.5684±0.0511
Secom	0.5502±0.0443	0.0871±0.0199	0.5596±0.0656	0.0932±0.0184
U2R	0.9836±0.0058	0.1858±0.0827	0.9881±0.0027	0.2982±0.1006

## Data Availability

No new data were created or analyzed in this study. Data sharing is not applicable to this article.

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
