# Peer review of "Unsupervised Anomaly Detection with Distillated Teacher-Student Network Ensemble"

_entropy, 2021, doi:10.3390/e23020201_

Round 1

Reviewer 1 Report

Dear authors,

in this work you discuss the problem of anomaly detection, proposing a novel approach that outperforms regarding 2 performance metrics several state-of-the-art algorithms. Your proposal combines a teacher-student model for recognizing instances for which some significant different behaviors are induced, assuming that this criterion can be utilized as an accurate anomaly detection solution. Although similar works can be found in the literature, you favorably mention your main novelties, which include the manipulation of categorical features on high-dimensional problems and the manner that you exploit several models and aspects from your data for boosting the total performance. The manuscript is well-written in general and the total structure of your work is well-designed and focused on commenting and discussing issues that are relevant to your work. I have some minor issues, mainly for improving the quality of your work, but i also highlight 2 major issues that have to be addressed.

Minor:

  1. Provide some references into the first paragraph, and a reference on lines 50-52. Please discuss also the approach of Catboost (CatBoost: unbiased boosting with categorical features)
  2. some simple grammatical errors (e.g. density- based, nomalities, produce na, by investigate). Consider also introducing the term pre-training. Rephrase line, 196 and 366. 
  3. Try to reduce the size of your figure captions. This should be at the most 2 lines! Include these descriptions into your main text.
  4. Algorithm 1 should be moved at the end of Section 4, augmented by the next modifications that you propose, so as to summarize end to end the proposed work.
  5. Use mAP abbreviation as you defined after its first met, as you make with other similar cases.
  6. Although you discuss in-depth your results providing insightful explanations and comments regarding the performed comparisons, you do not sum up smooth through the last Section, providing a restricted discussion there. Try either to merge the last two Sections or provide some future works (+references) regarding your proposed work, discussing for example some more soft-computing approaches that could be utilized, or other criteria that are based for example on diversity measurements for getting more informative training samples for the student models.

As it concerns the major issues: 

  1. There are some points that are ambiguous and should be discussed further. What happens with the λ and α parameters? How did you finally set them? Dynamically or empirically? Can you provide any investigating experimental result as the study about the random self-supervised variant of the proposed algorithm? Moreover, you mention that multiple anomaly scores are computed. I would anticipate to depict these scores and understand better their utility.
  2. Moreover, you mention that you address high-dimensional categorical feature spaces, but Table 1 does not mention how many of the features are categorical. Do these datasets contain both numerical and categorical features? Please discuss and think again your main contributions of this does not actually fit on your case.
  3. Perform a statistical comparison on your results, for examining the statistical significance of the obtained performances.

Reviewer 2 Report

Paper deals with very important task. The proposed unsupervised approach for anomaly detection of multivariate categorical data. Experimental investigation was conducted using 10 datasets. It is very good.

Paper has a good structure, scientific novelty and great practical value.

Suggestions:

  1. Introduction section should be extended. Authors should take into account ANN-based non-iterative approach to training in supervised and unsupervised modes. They can use this paper DOI: 10.1007/978-3-319-91008-6_58 among others.
  2. Fig. 1, 3, 4 are very small. Please fix it
  3. Please provide PC parameters, used for modeling
  4. Please provide the information about optimal parameters for all anomaly detection methods used for modeling.
  5. Conclusion section should be significantly extended using the limitations of the proposed approach and directions for the future research.
  6. Some references are outdated. Please fix it.

Round 2

Reviewer 1 Report

Dear authors,

thank you for your in-depth analysis of the mentioned issues.

Almost all the issues have been addressed properly.

I highlight the next ones:

i) use of pre-trained instead of pretrained across all the manuscript

ii) Great visualization of the statistical testing. Please report the significance level (a = 0.05?) and the corresponding Critical Difference (CD), since you have only drawn it on Figure 3, as well as mention which post-hoc test was used for computing this value.

iii) Figures 6 and 7 could be better recorded into matrices.

iv) Elaborate the future work paragraph in the last Section (it seems awkward).

I also suggest sharing your implementation, for increasing your work soundness.

In total, I believe that this work satisfies now the criteria for publication.